biotechnology, microbiology, ecology

rhizosphere microbiome, biodiversity, probiotic consortia, multifunctionality, *Pseudomonas*, plant growth promotion

**Author for correspondence:**
Zhong Wei
e-mail: weizhong@njau.edu.cn

†These authors contribute equally to this work.

# Introduction of probiotic bacterial consortia promotes plant growth via impacts on the resident rhizosphere microbiome

Jie Hu[1,2,†], Tianjie Yang[1,†], Ville-Petri Friman[3], George A. Kowalchuk[2], Yann Hautier[2], Mei Li[1,2], Zhong Wei[1], Yangchun Xu[1], Qirong Shen[1] and Alexandre Jousset[1,2]

[1]Jiangsu Provincial Key Lab for Organic Solid Waste Utilization, Key Lab of Plant immunity, National Engineering Research Center for Organic-based Fertilizers, Jiangsu Collaborative Innovation Center for Solid Organic Waste Resource Utilization, Nanjing Agricultural University, Weigang 1, Nanjing 210095, People's Republic of China
[2]Institute for Environmental Biology, Ecology and Biodiversity, Utrecht University, Padualaan 8, Utrecht 3584CH, The Netherlands
[3]Department of Biology, University of York, Wentworth Way, York YO10 5DD, UK

 JH, 0000-0002-0965-8197; TY, 0000-0003-0927-2670; V-PF, 0000-0002-1592-157X; ZW, 0000-0002-7967-4897; QS, 0000-0002-5662-9620; AJ, 0000-0002-6805-2486

Plant growth depends on a range of functions provided by their associated rhizosphere microbiome, including nutrient mineralization, hormone co-regulation and pathogen suppression. Improving the ability of plant-associated microbiomes to deliver these functions is thus important for developing robust and sustainable crop production. However, it is yet unclear how beneficial effects of probiotic microbial inoculants can be optimized and how their effects are mediated. Here, we sought to enhance tomato plant growth by targeted introduction of probiotic bacterial consortia consisting of up to eight plant-associated *Pseudomonas* strains. We found that the effect of probiotic consortium inoculation was richness-dependent: consortia that contained more *Pseudomonas* strains reached higher densities in the tomato rhizosphere and had clearer beneficial effects on multiple plant growth characteristics. Crucially, these effects were best explained by changes in the resident community diversity, composition and increase in the relative abundance of initially rare taxa, instead of introduction of plant-beneficial traits into the existing community along with probiotic consortia. Together, our results suggest that beneficial effects of microbial introductions can be driven indirectly through effects on the diversity and composition of the resident plant rhizosphere microbiome.

## 1. Introduction

Microorganisms associated with plant roots provide a range of services essential for plant growth. Many species mineralize nutrients, produce growth hormones or prevent diseases [1,2]. However, while many functions have been described, it has remained challenging to harness the multiple functions encoded in the microbiome. This is particularly an issue in soils degraded by intensive agriculture. In such soils, low microbial biodiversity restricts the number of plant-beneficial functions and makes their expression by microbes unstable [3]. Consequently, soil fertility may be improved by restoring microbial biodiversity, thereby re-establishing community-level delivery of the multiple plant-beneficial services needed for a robust growth [4,5].

One way to increase microbiome-associated multifunctionality is the introduction of additional microbes into the soil [6]. For a long time, microorganisms

have been selected on the basis of their ability to directly express functions of interest [7], such as nutrient mineralization, nitrogen fixation or pathogen suppression [8]. However, such direct effects are notoriously unstable as the introduced species fail to establish at the density needed to function in a natural microbiome context [9]. Further, inherent trade-offs in microbial physiology will limit the expression of functions one microbial species or strain can provide to the plant [10]. Such limitations could be partly solved by using multispecies probiotic consortia to improve both the inoculant establishment and the variety of services microbes can provide to the plant [11,12] compared to single-species inoculants [13]. For example, a consortium of multiple species that do not show antagonism towards each other may together occupy a broader range of ecological niches [14], allowing them to better colonize plant rhizosphere. Furthermore, a more diverse microbial consortium is likely to contain a larger amount of plant-beneficial functions [11,14,15], increasing the consortium-level functional diversity and redundancy. Hence, application of microbial consortia could potentially introduce multiple plant-beneficial functions into the soil that could be expressed simultaneously by different consortium members due to ecological complementarity.

In addition to directly introducing functions to the rhizosphere, inoculated microbes could have indirect effects via alteration of the diversity, composition and functioning of the resident rhizosphere microbiome [16,17]. Previous studies have shown that microbial invasions can considerably shift microbiome diversity, composition and functioning, leading to significant effects even over successive plant generations [4,5]. Such changes could be mediated via microbial competition for resources or antagonism triggered by antibiosis [18], which could change the balance between dominant and rare taxa [19], indirectly driving pathogen suppression. Some inoculated species could also interact with the plant, triggering plant-mediated 'steering' of microbiomes via altering root exudation patterns [20], producing plant-derived antimicrobials [21] or inducing other plant defences [22]. If the effects of species introductions are diversity-dependent [23], diverse microbial inoculants could trigger relatively larger shifts in the functioning of the resident microbiome [24]. While such potential indirect effects of microbial inoculants on the functioning of the resident rhizosphere microbiome have been reported previously [16,17], they are still relatively poorly understood.

In this study, we employed a biodiversity-ecosystem functioning framework [25] to assess how the diverse probiotic bacterial inoculants affect rhizosphere microbiome and their impact on plant growth. To this end, we assembled probiotic bacterial consortia consisting of one to eight different *Pseudomonas* spp. strains that all are well studied and have previously been shown to have beneficial effects on plants (electronic supplementary material, tables S1–S3) [26]. We first characterized the *in vitro* performance of each consortium regarding bacterial traits linked to plant-beneficial functions (electronic supplementary material, table S3). The mean effects of individual plant-beneficial functions were further analysed by a weighted multifunctionality index [27], which summarized the overall consortium ability to provide different functions simultaneously. We then inoculated each combination of probiotic consortia in the tomato rhizosphere and analysed the consortium effects and the contribution of each *Pseudomonas* strain present in consortia (strain identity effects). First, we measured the consortium effects on plant growth, nutrient assimilation and protection

against the soil-borne pathogen *Ralstonia solanacearum* [28]. Second, we used 16S rRNA amplicon sequencing to obtain a snapshot of the tomato rhizosphere microbiome after the inoculation of *Pseudomonas* consortia, and assessed the probiotic inoculant colonization. Finally, structural equation modelling (SEM) was used to compare the contribution of direct and indirect effects of the inoculated consortia on plants. We hypothesized that introduction of *Pseudomonas* consortia could promote plant growth, and that this promotion could be magnified with increasing consortium richness. Mechanistically, such effects could be driven by improved *Pseudomonas* establishment and introduce novel functional traits into the microbiome directly, or indirectly via changes on the composition and functioning of the existing resident microbiome.

## 2. Material and methods

### (a) Model bacterial strains

We used eight strains of *Pseudomonas* spp. as model organisms: *P. fluorescens* 1m1–96, *P. fluorescens* mvp1–4, *P. fluorescens* Phl1c2, *P. fluorescens* Q2–87; *P. kilonensis* F113, *P. protegens* Pf-5, *P. protegens* CHA0 and *P. brassicacearum* Q8r1-96 (electronic supplementary material, table S1). These strains are well-characterized model biocontrol strains and widely used in plant growth promotion and pathogen-suppression studies [15,26,29]. They express different traits that are important for several microbiome functions that improve plant growth (electronic supplementary material, table S3). All these strains contain functional gene *phlD*, which is involved in the production of antimicrobial polyketide 2,4-diacetylphloroglucinol (DAPG). This gene was used as a molecular marker to quantify the abundances of introduced *Pseudomonas* strains, as the background density of this gene was very low in the used soil (below 0.001% relative abundance in the non-inoculated control treatment; (electronic supplementary material, figure S4A). *Ralstonia solanacearum* QL-Rs1115 strain was used as a model soil-borne pathogen in *in vitro* and *in vivo* experiments. All strains were store at −80°C, and prior to all experiments, one single colony of each bacterial strain was selected and prepared by grown overnight in nutrient broth medium, washed three times in 0.85% NaCl buffer and adjusted to a density of $10^8$ cells ml$^{-1}$.

### (b) Assembly of probiotic *Pseudomonas* consortia

We assembled a total of 37 distinct *Pseudomonas* consortia in 48 treatments that contained 1, 2, 4 or 8 *Pseudomonas* spp. strains (four richness levels) following a substitutive design (electronic supplementary material, table S2) [15,30,31]. In this design, each strain is equally often present in different consortia within each richness level, allowing discriminating richness effects from confounding species identity effects later in the analysis [31]. Moreover, bacterial suspensions of individual *Pseudomonas* species were mixed in equal amounts in all consortia to keep the total bacterial abundances the same between different richness levels and immediately used for subsequent experiments. The same substitutive experimental design was used in both the *in vitro* assays and *in vivo* greenhouse experiments.

### (c) Measurement of *in vitro* functional traits of the assembled consortia and calculation of multifunctionality index

The original data related to *in vitro* functional traits of the assembled consortia were measured in previously published

studies [11,15]. In this study, all previously collected data were combined for an integrated meta-analysis (electronic supplementary material, table S4) and short description of measurement details are in the electronic supplementary material.

## (d) Calculation of *in vitro* multifunctionality index

All measured *in vitro Pseudomonas* traits were used to compute a multifunctionality index for each consortium using a weighted average standardized calculation method [27]. Briefly, all measured variables were first correlated with consortium richness to detect the direction of correlations, then standardized between 0 and 1 by using a modified function of the getStdAndMeanFunctions in R [32]. To avoid an overrepresentation of functions with similar contribution to overall ecosystem functioning, we applied a cluster analysis [27] to identify closely related functions. We assigned a weight of one to each cluster and weighed each function equally within each cluster so that functions in a cluster summed to one. The weighted average multifunctionality was then calculated based on the dendrogram. For example, siderophore and auxin production were assigned to one cluster and each of them received a weight of 0.5 (electronic supplementary material, figure S1A).

## (e) Greenhouse experiments

We assessed the effect of consortium richness on microbiome diversity and function in two separate greenhouse experiments. In both experiments, we grew tomato plants in non-sterile agricultural soil collected from a tomato field in Qilin town of Nanjing, China, which has been used for tunnel greenhouse farming for over 15 years [33]. Surface-sterilized tomato seeds (*Lycopersicon esculentum*, cultivar 'Jiangshu') were germinated on water-agar plates for 3 days before sowing into seedling plates containing $^{60}$Co-sterilized seedling substrate (Huainong, Huaian Soil and Fertilizer Institute, Huai'an, China). Germinated tomato plants were transplanted to seedling trays containing non-sterile agricultural soil at the three-leaf stage (12 days after sowing). Two tomato plants per cell (600 g of soil) were transplanted to total of eight separate cells resulting in sixteen seedlings per seedling tray (370 × 272 × 83 mm). Each tray was treated as one biological replicate resulting to a total of 104 trays. Half of these trays ($n = 52$) were used for the plant growth experiment, and the other half ($n = 52$) for the plant protection experiment; more detail included in the electronic supplementary material.

## (f) Calculation of weighted average plant growth index

All measured plant traits (plant aboveground dry biomass; nitrogen, potassium, phosphorus and iron concentrations in the plant tissue; and plant disease severity index) were used to calculate weighted average plant growth index [27]. To avoid overrepresentation of functions with similar contribution, we applied the cluster analysis to all measured traits, similar to when previously calculating the *in vitro Pseudomonas* consortium multifunctionality index. For example, aboveground plant biomass and protection against pathogen were assigned to one cluster and each of them received a weigh of 0.5 (electronic supplementary material, figure S1B).

## (g) DNA extraction, qPCR quantification and sequencing
### (i) Soil DNA extraction

Rhizosphere soil from the plant growth promotion experiment was collected by gently removing plants from the trays before shaking off excess soil and collecting the soil attached to the root system. Two randomly selected plants from different cells per seedling tray were collected and pooled together. Samples were then suspended into 30 ml of sterile $H_2O$ (100 r.p.m., 30 min at 4°C) and centrifuged (5000$g$, 30 min at 4°C) before

the soil pellets were transferred into 2 ml tubes and stored at −80°C for subsequent experiments. We extracted DNA with the Power Soil DNA Isolation Kit (Mobio Laboratories, Carlsbad, CA, USA). Briefly, DNA from 0.3 g of soil pellets per sample was extracted following the manufacturer's protocol. DNA fragment size was checked on 1% agarose gel, and DNA concentration and purity were determined with a NanoDrop 1000 spectrophotometer (Thermo Scientific, Waltham, MA, USA) prior to downstream analyses.

### (ii) Quantification of total bacterial and *Pseudomonas* abundances

We used quantitative PCR (qPCR) to quantify the abundance of total bacteria associated with plant rhizosphere soil as well as the introduced *Pseudomonas* consortia based on 16S rRNA and *phlD* gene copies per gram of soil, respectively; more detail included in the electronic supplementary material, methods.

### (iii) 16S rRNA amplicon sequencing

We used the 563F (5′-AYT GGG YDT AAA GVG-3′) and 802R (5′-TAC NVG GGT ATC TAA TCC-3′) [34] primer pair to amplify the V4 hypervariable region of the bacterial 16S rRNA gene; more detail included in the electronic supplementary material, methods. Raw fastq files were demultiplexed and quality-filtered with QIIME (v. 1.17) [35] according to previous established protocols [17]. After discarding unqualified reads, the operational taxonomic units (OTUs) were assigned at 97% identity level using UPARSE [36] and chimeric sequences identified and removed using UCHIME [37]. The phylogenetic affiliation of each 16S rRNA gene sequence was analysed using the RDP Classifier [38] against the Silva 16S rRNA gene database with a confidence threshold of 70% [39]. We removed unassigned, Archaea, mitochondrial and plastid OTUs and those found in fewer than 3 times in less than 1% of the samples, more detailed description was in supplementary materials. PyNast and FastTree were used to estimate the phylogeny of all OTUs observed in all samples. To obtain an equivalent sequencing depth for later analysis, the sample OTU abundances were rarefied by using the lowest sample sequence depth, resulting in similar sequence depths between all samples (mean = 29 068, min = 28 749, max = 29 465). To ensure the robustness of the OTU bioinformatic pipeline, the 16S rRNA Miseq sequence data was also analysed using DADA2 pipeline (more detail is included in the electronic supplementary material, methods; see electronic supplementary material, figure S2 for a reproduction of figure 4*a* and electronic supplementary material, figure S3 for a reproduction of figure 4*c* with DADA2 pipeline).

### (iv) Calculation of resident microbiome biodiversity

We characterized the biodiversity of resident bacterial communities using phylogenetic abundance evenness (PAE), an index accounting for both evenness and phylogenetic distribution [40] (more detail is included in the electronic supplementary material).

## (h) Statistical analyses

To analyse the effect of richness and abundance of probiotic *Pseudomonas* consortia on the resident rhizosphere microbiome community composition, diversity and weighted mean of plant growth index, generalized linear models (GLMs) were used. Consortium richness was treated as a factor, *phlD* abundance in rhizosphere was log-transformed, and their effects on subsequently measured parameters were assessed with GLMs using a Gaussian distribution in two models with different factor directions following a subsequent ANOVA analysis in R. The effect of microbial inoculation on rhizosphere microbiome community composition was determined with a redundancy analysis (RDA)

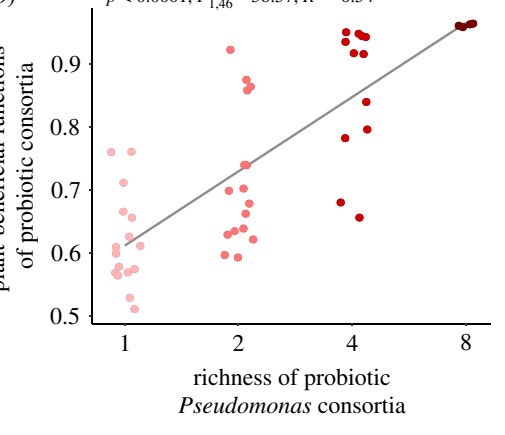

(a)

- Pf–5
- 1m1–96
- mvp1–4
- F113
- Q8r1–96
- Phl1c2
- Q2–87
- CHA0

- T1: carbon catabolism
- T2: siderophore production
- T3: auxin production
- T4: phosphate solubilizaton
- T5: gibberellin production
- T6: antibacterial activity

(b) $p < 0.0001, F_{1,46} = 56.57, R^2 = 0.54$

plant-beneficial functions of probiotic consortia

richness of probiotic *Pseudomonas* consortia

**Figure 1.** *In vitro* plant-beneficial functions provided by the probiotic *Pseudomonas* strains and consortia. (a) Radar chart showing the mean ability of individual *Pseudomonas* strains of to express six plant-beneficial functions measured *in vitro*. (b) Positive correlation between the richness of probiotic *Pseudomonas* consortia and mean of all plant-beneficial functions measured *in vitro* at the consortium level (consortia assembled based on substitutive design presented in electronic supplementary material, table S2). (Online version in colour.)

based on the taxa presence–absence OTU table or a principal coordinates analysis (PCoA) based on the weighted Bray–Curtis distance metric, consortium richness (factor with five levels) and *phlD* gene copy numbers in the rhizosphere, using the R function vegan: rda. Coordinates were used to draw two-dimensional graphical outputs, and the effect of consortium richness and establishment success on rhizosphere microbiome community composition was tested using the R function vegan: adonis. To explore the significance of *Pseudomonas* strain properties for the shift in resident microbiome, the effect of *in vitro* plant-beneficial functions measured at the consortium level were used to explain variation in the resident rhizosphere microbiome composition, which was presented by using canonical correlation analysis (CCA). Stepwise GLMs were used to test the effect of *in vitro* plant-beneficial functions, such as breadth of carbon catabolism, antibacterial activity, phytohormone production and nutrient availability, on resident microbiome composition. Finally, we used SEM to examine direct and indirect effects linking consortium inoculants with plant growth by accounting for multiple potentially correlated effect pathways [41] (more detail is included in the electronic supplementary material).

## 3. Results

### (a) Effect of richness and *Pseudomonas* strain identities on probiotic consortium plant-beneficial functions

Each of the studied *Pseudomonas* strains excelled at expressing a specific subset of plant-beneficial functions, including ability to consume different carbon resources, production of auxin, gibberellin, siderophores or antibacterial activity (figure 1a). For example, *P. protegens* Pf-5 produced the highest amount of siderophores but the lowest levels of auxin (figure 1a). By contrast, *P. fluorescens* 1M1-96 produced the highest levels of auxin but had a very weak antibacterial activity (figure 1a). These differences suggest that *Pseudomonas* strains were specialized in different plant-beneficial functions and could potentially show complementary effects when combined in multi-strain consortia. To test this, we used weighted means of all the individual plant-beneficial functions and combined them into predicted consortium multifunctionality index. It was found that consortium richness correlated positively with the consortium multifunctionality index (figure 1b; $F_{1,46} = 60.02$, $p < 0.0001$), and increasing the consortium

richness promoted the expression of all the individual plant-beneficial functions when measured directly *in vitro* (electronic supplementary material, table S5). Some of the individual functions were further influenced by the presence of certain strains. For instance, consortia containing the *P. protegens* CHA0 and Pf-5 strains showed the highest levels of antibacterial activity (electronic supplementary material, table S5). However, despite certain strong identity effects, consortium richness remained a major significant predictor even after accounting for the inclusion of each strain.

### (b) Effect of consortium richness and *Pseudomonas* strain identities on plant growth characteristics

Similar to *in vitro* measurement of plant-beneficial functions, each *Pseudomonas* strain had unique impacts on plant growth on their own. For example, *P. kilonensis* F113 inoculation led to a high aboveground plant biomass, *P. protegens* CHA0 inoculation led to a high concentration of potassium and phosphorous in plant tissues, while *P. brassicacearum* Q8r1-96 inoculation led to the highest protection against *R. solanacearum* pathogen (figure 2a). Single *Pseudomonas* strain inoculations affected only a limited number of plant growth characteristics (figure 2a,b), increasing consortium richness led to an improvement of multiple plant growth characteristics (figure 2b). All individual plant growth characteristics were positively correlated with consortium richness except for plant tissue nitrogen concentration (figure 2b). As a result, the weighted average plant growth index also correlated positively with the probiotic consortium richness (figure 2b). By contrast to the *in vitro* plant-beneficial functions expressed by consortia, the *Pseudomonas* strains had no significant identity effects on the plant growth (figure 2; electronic supplementary material, table S5).

### (c) Effect of richness and *Pseudomonas* strain identities on consortium establishment in the rhizosphere

The background *phlD* gene abundances were below 0.001% (relative to 16S rRNA gene abundances) in the non-inoculated control treatments (electronic supplementary material, figure S4A), confirming that *Pseudomonas* strains containing this gene were rare in the non-sterile agricultural soil

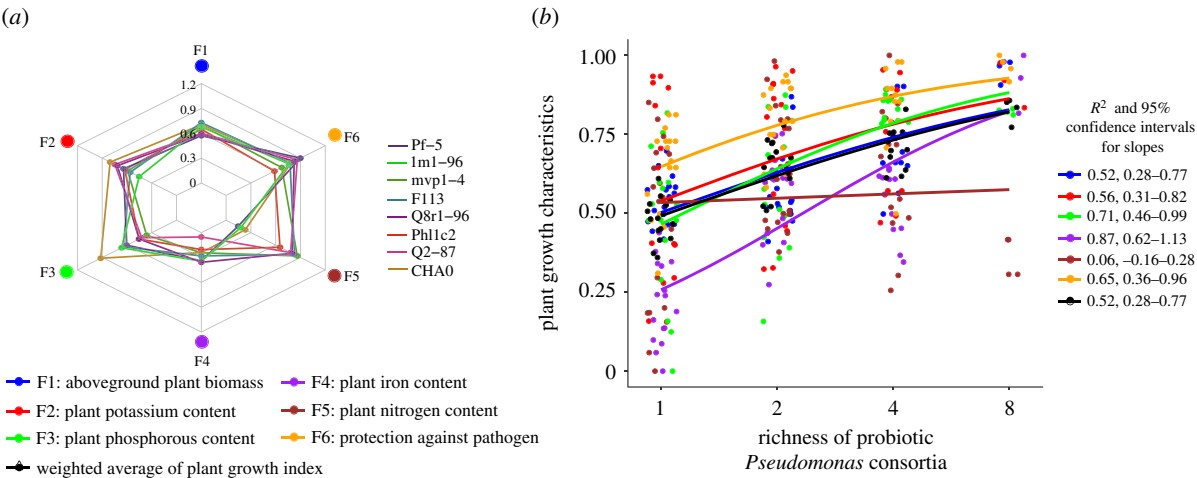

**Figure 2.** Plant growth benefits provided by the introduced *Pseudomonas* strains and consortia measured *in vivo*. (*a*) Radar chart showing the mean effect of individual *Pseudomonas* strains on six plant growth characteristics. (*b*) Correlation between the richness of probiotic *Pseudomonas* consortia and the six individual plant growth characteristics (coloured lines), and the weighted average of plant growth index (black line) by probiotic consortia (consortia assembled based on substitutive design presented in electronic supplementary material, table S2; each point shows the mean effects of three replicate plants ($n = 3$) for each plant growth characteristic when exposed to different consortia). The colours in the key of (*a*) correspond to the same growth characteristics and colours presented in (*b*). (Online version in colour.)

**Table 1.** Effect of richness and abundance of probiotic *Pseudomonas* consortia on the resident rhizosphere microbiome community composition, diversity and weighted average of plant growth index. Two-directional Adonis model was used for testing the effect of probiotic consortium richness and abundance on the resident microbiome composition, while general linear models were used to test the effect on resident microbiome diversity and weighted average of plant growth index. Significant results ($p < 0.05$) are highlighted in italics. Upward arrows denote positive effects of retained explanatory variables in all the models and '1st' and '2nd' indicate the order of each parameter when fitting the model.

| variable | d.f. | resident microbiome composition | | resident microbiome diversity | | weighted average of plant growth index | |
|---|---|---|---|---|---|---|---|
| | | *F*. model | *p* | *F* | *p* | *F* | *p* |
| richness of probiotic consortia | | | | | | | |
| 1st | 1 | 1.06 | *0.0280*↑ | 4.87 | *0.0324*↑ | 60.02 | *<0.0001*↑ |
| 2nd | 1 | 0.99 | 0.4730 | 3.93 | 0.0535 | 0.66 | *0.0106*↑ |
| abundance of the probiotic consortia | | | | | | | |
| 1st | 1 | 1.00 | 0.4620 | 1.11 | 0.2987 | 29.58 | *<0.0001*↑ |
| 2nd | 1 | 1.06 | *0.0250*↑ | 0.16 | 0.6883 | 31.10 | 0.1119 |
| no. of residuals | 45 | | | | | | |
| AIC | | — | | −179.60 | | −88.90 | |

used in the pot experiments. By contrast, *phlD* gene abundances ranged between 0.01% and 0.25% of the total 16S rRNA gene abundances in samples with inoculated *Pseudomonas* strains. Specifically, we found that *phlD* gene abundances increased along with consortium richness gradient (electronic supplementary material, figure S4A) and only mvp1–4 strain had a marginally significant positive identity effect on the relative *phlD* gene abundances ($p = 0.04$, electronic supplementary material, table S6).

## (d) Effect of consortium richness on the abundance, composition and diversity of the resident rhizosphere microbiome

We found that irrespective of inoculant richness, all probiotic consortia had similar negative effects on the abundance of resident rhizosphere bacteria compared to a non-inoculated control treatment (electronic supplementary material, figure S4B). All inoculated consortia changed the composition of the resident rhizosphere microbiome. However, this impact was magnified with increasing consortium richness ($p = 0.0280$; table 1; figure 3*a*), qualitatively similar results ($p = 0.0440$; electronic supplementary material, figure S4C) were also obtained using weighted Bray–Curtis distance metric. Interestingly, a clear increase in resident microbiome diversity was observed along with consortium richness (table 1; electronic supplementary material, table S7), even though no differences between single- and two-strain consortia, or four- and eight-strain consortium inoculants, were found (figure 3*b*). Moreover, the effects of consortium richness on microbiome diversity remained significant even after accounting for *Pseudomonas* strain identity effects (electronic supplementary material, table S7). One explanation for

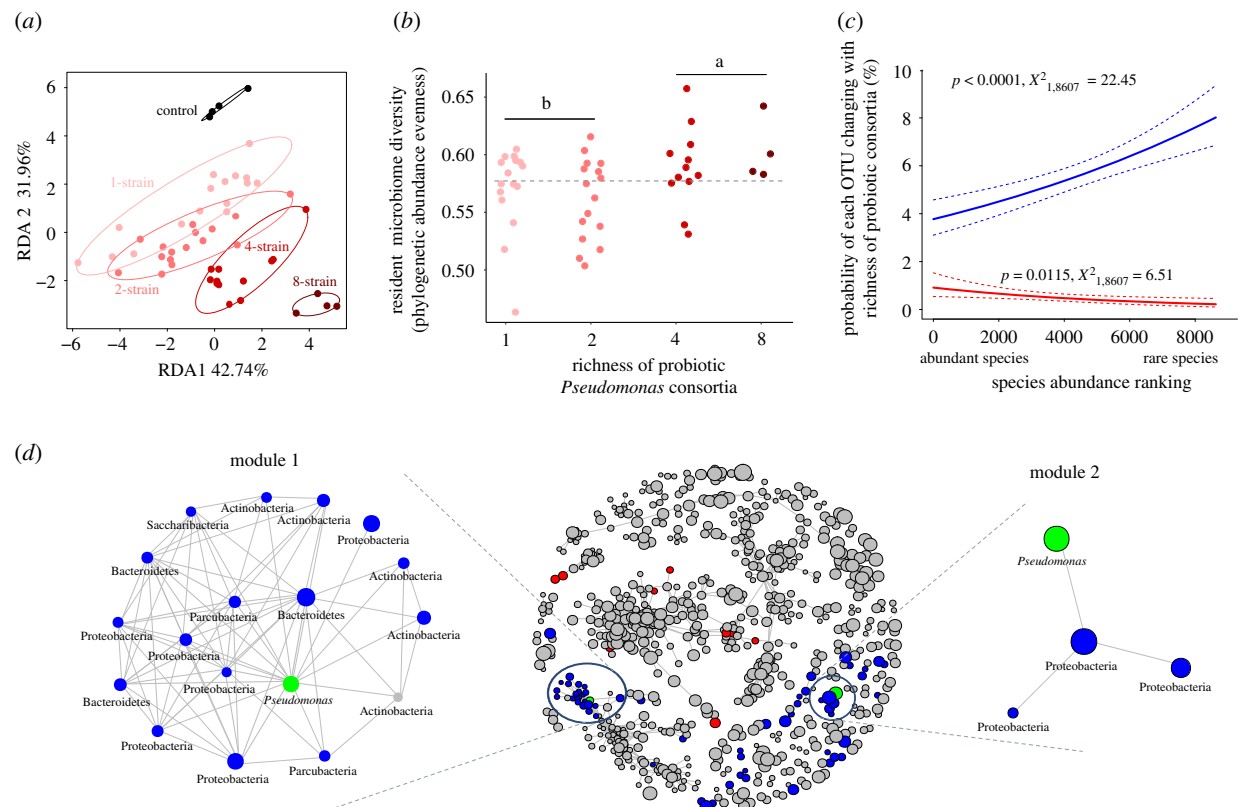

**Figure 3.** Effects of probiotic *Pseudomonas* consortia on the composition and diversity of the resident rhizosphere microbiome. (*a*) RDA analysis of resident microbiome composition showing the impact of increasing consortium richness on the rhizosphere microbiome composition. Ellipses enclosing points at each richness level show 95% confidence intervals. (*b*) Effect of probiotic consortium richness on the rhizosphere microbiome diversity in terms of phylogenetic abundance evenness. The grey dashed line shows the mean diversity in the control treatment (non-sterile agricultural field soil without introduced consortia). Tukey HSD analysis was performed to compare low-diversity (one- and two-strain) and high-diversity (four- and eight-strain) consortia. In both (*a*) and (*b*), each point corresponds to different consortium (based on substitutive design presented in electronic supplementary material, table S2). (*c*) Response of abundant and rare rhizosphere microbiome taxa to increasing richness of probiotic consortia. The blue line shows the binomial regression on the probability of a given OTU (*y*-axis), which was initially abundant or rare (*x*-axis, OTUs were ranked based on their relative abundance in the control treatments), to increase in its relative abundance along with consortium richness. The red line shows the binomial regression on the probability of a given OTU (*y*-axis), which was initially abundant or rare (*x*-axis), to decrease in its relative abundance along with consortium richness. Dashed lines show 95% confidence interval of each binomial regression. (*d*) Co-occurrence network of the resident microbiome (in the middle), with each node representing a bacterial OTU where the node size is proportional to relative OTU abundance. Blue and red circles represent OTUs that were significantly positively and negatively correlated with consortium richness, respectively. Green circles indicate OTUs (OTU 8478 in module 1 and OTU4923 in module 2) putatively belonging to the inoculated CHA0 and mvp1–4 strains. Links between nodes show statistically significant positive Spearman correlations ( *p* < 0.05) with correlation coefficient greater than 0.8 (no significant negative correlations found). Left and right insets show detailed overview of modules 1 and 2, highlighting the associations between resident taxa and putative probiotic *Pseudomonas* OTUs (OTUs are shown at lowest taxonomic level that could be assigned by the bioinformatic analysis, ranging from phylum to genus level). (Online version in colour.)

the increase in the resident community diversity is that *Pseudomonas* inoculants potentially indirectly favoured rare species by competing more intensively with relatively abundant taxa. In support for this, we found that rare taxa at phylum, family and OTU levels were more likely to increase in abundance along with the richness of the inoculated consortia ($\chi^2_{1,8607} = 22.45$, $p < 0.0001$; figure 3*c*; electronic supplementary material, figure S5A–C). By contrast, the most abundant taxa were more likely to decrease in response to *Pseudomonas* inoculations ($\chi^2_{1,8607} = 6.51$, $p < 0.0001$; figure 3*c*; electronic supplementary material, figure S5D). At a coarser taxonomic level, we observed that resident microbiome responses to *Pseudomonas* inoculants were conserved with some phyla. For instance, OTUs belonging to the phyla Candidate division WS6, Thermotogae, Spirochaetae, Gracilibacteria and Parcubacteria consistently increased along with the *Pseudomonas* consortium richness (electronic supplementary material, figure S5A, table S8). Finally, we conducted covariance network analysis to assess whether the introduced *Pseudomonas*

strains were embedded in the resident microbiome. Despite the limited taxonomic resolution of amplicon sequencing, we could identify two *Pseudomonas* sp. OTUs (shown as green circles in modules 1 and 2, figure 3*d*) putatively matching with two introduced *Pseudomonas* strains (CHA0 and mvp1–4 strains). These OTUs covaried positively with a range of other OTUs present in the resident rhizosphere microbiome, including Parcubacteria, Proteobacteria, Actinobacteria, Bacteroidetes and Saccharibacteria phyla (figure 3*d*; electronic supplementary material, table S9).

### (e) Disentangling direct versus indirect effects of consortium richness on plant growth

We used SEM to disentangle whether the effects of probiotic consortia on the plant growth were directly driven by the plant-beneficial functions introduced by consortia, or indirectly via shifts in the resident rhizosphere microbiome. We found that plant-beneficial functions of consortia measured

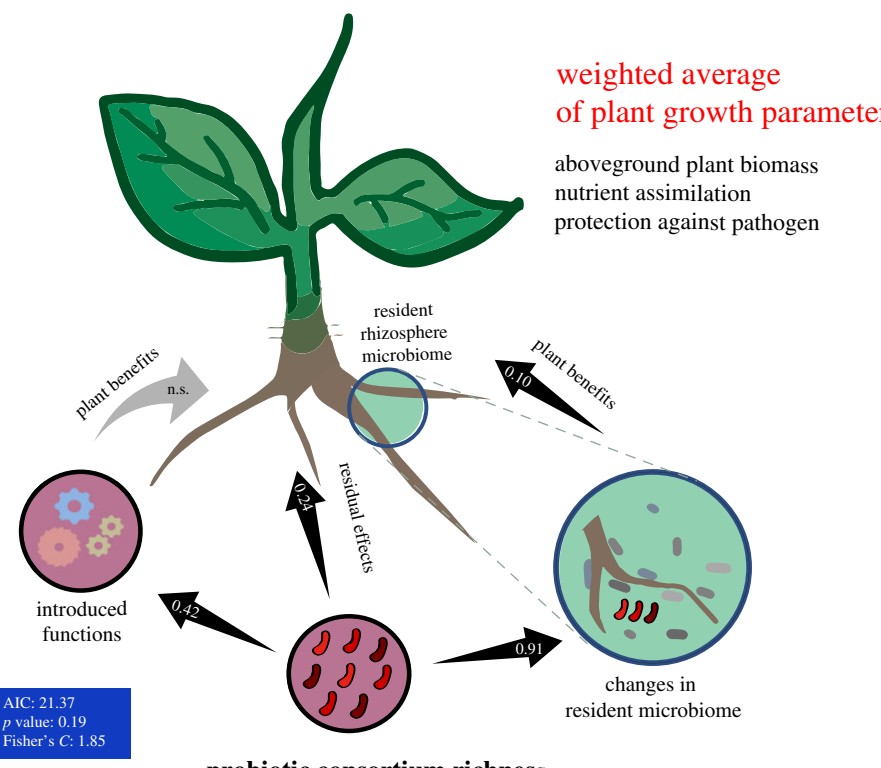

**Figure 4.** Structural equation model (SEM) comparing the direct and indirect effects of consortium richness on plant growth (weighted average of plant growth parameters). Black arrows indicate significant relationships between the tested variables, while grey arrows indicate non-significant relationships retained in the model. 'Introduced functions' refer to the consortium multifunctionality index, while 'changes in resident microbiome' refers to shift in the resident microbiome composition after inoculation (the first RDA axis of multivariate community composition analysis). The numbers inside the arrows indicate standardized correlation coefficients (relative effect sizes of non-significant correlations are not shown). The parameters inside the blue rectangle on the left show the robustness of the SEM model. (Online version in colour.)

*in vitro* predicted poorly the individual plant growth characteristics (electronic supplementary material, figure S6A–C). The only notable exception was a positive relationship between the consortium phosphate solubilization capacity and the plant shoot phosphorus content (electronic supplementary material, figure S6D). Similarly, the multifunctionality index of probiotic consortia did not explain the positive effects of inoculants on the plant growth (left side, figure 4). Instead, *Pseudomonas* consortium richness had strong effects on the composition of resident plant microbiome, which in turn was positively correlated with weighted average plant growth index (figure 4). Interestingly, consortium diversity also had a direct positive effect on weighted average plant growth index (residuals; figure 4).

To gain additional insight into underlying mechanisms, we used CCA to describe the resident microbiome composition and assessed if *Pseudomonas* consortium traits measured *in vitro* explained the resident microbiome composition. We found that the variation on CCA1 was best explained by the *in vitro* production of auxin and gibberellin phytohormones by the introduced consortia (electronic supplementary material, table S10). By contrast, variation on the CCA2 was best explained by the niche breadth of the probiotic consortia (i.e. their ability to metabolize an array of carbon sources typically found in the rhizosphere). As access to resources is crucial for inoculant establishment, these data suggest that probiotic consortia could have shaped the resident microbiome through resource competition. By contrast, the antibacterial activity of *Pseudomonas* consortia was not retained in the final model (electronic supplementary material, table S10).

## 4. Discussion

Here we used a biodiversity–ecosystem functioning framework to directly assess how the diversity of probiotic bacterial inoculants affect microbiome structure and plant growth. Specifically, we explored if the potential benefits were driven directly by introduced consortia via introduction of essential functions to the microbiome, or indirectly via changes in the resident bacterial community. It was found that increasing probiotic consortium richness increased inoculant colonization success, and was associated with improved plant growth, nutrient assimilation and protection from pathogen infection. Crucially, inoculants caused shifts in the resident microbiome and these effects were magnified with increasing probiotic consortium richness, leading to an increase in the abundance of rare taxa and overall microbiome biodiversity. While some significant *Pseudomonas* strain identity effects were found, the improvement of plant growth was poorly explained by plant-beneficial functions provided by different consortia members. Instead, positive effects on the plant growth were best explained by consortium-mediated shifts in the resident microbiome, which were associated with phytohormone production and resource competition by the probiotic consortia. Together these findings suggest that probiotic bacteria can be used to steer the existing resident rhizosphere microbiome in order to improve plant growth.

Multi-strain consortia could perform better together due to ecological complementarity between plant-beneficial functions they can provide at consortium level, or due to other emergent diversity effects arising in complex microbe–

microbe–plant communities [18]. While the probiotic consortium richness correlated positively with its multifunctionality and several plant growth characteristics, the consortium multifunctionality poorly predicted plant growth. Instead, positive diversity effects via shifts in the resident microbiome and the probiotic consortium richness itself were better predictors. Overall, probiotic consortium richness effects could be explained by improved establishment success in terms of relative *Pseudomonas* abundances in the rhizosphere. While certain *Pseudomonas* strains had strong identity effects, the effect of consortium richness remained significant even after their removal. Moreover, the densities of four- and eight-strain consortia were up to ten times higher than the best-performing single-strain inoculants, which indicates that richness effects were driven by emergent consortium level effects, making it ultimately difficult to disentangle the relative importance of density versus diversity effects. Such unexpectedly strong community performance is called transgressive overyielding [42], and could be indicative of synergistic interactions between inoculated *Pseudomonas* strains and the resident microbiome or the plant. The fact that inoculation of probiotic bacteria resulted in reduction in the abundance of resident bacteria suggests that *Pseudomonas* bacteria probably promoted competition with the resident community. In support for this, the consortium resource niche breadth increased along with the richness gradient, which might have allowed strains to sequester nutrients more efficiently in the rhizosphere [43]. However, the reduction in the resident bacteria abundances was the same for all consortia, and hence, resource competition alone is unlikely to explain the observed relationship with *Pseudomonas* consortium richness.

Interestingly, probiotic consortium richness and establishment success were positively associated with relatively larger and non-random changes in the diversity and composition of resident rhizosphere microbiome, with four- and eight-strain consortia showing the clearest effects. Specifically, diverse probiotic consortia were associated with more even resident bacterial communities and clearer increase in the abundance of rare bacterial taxa. One potential explanation for this could be inhibition of dominant organisms via production of secondary metabolites by the introduced consortia [44]. Alternatively, it is possible that *Pseudomonas* consortia reduced competitive exclusion of rare species by having disproportionally stronger competitive effect on the more abundant taxa. While additional work is required to unravel exact mechanisms, increase in the relative abundance of rare taxa was associated with decrease in the abundance of dominant taxa (figure 3c), indicative of highly asymmetric responses by different resident microbiome taxa. For example, four- and eight-strain probiotic consortia had proportionally large, positive effects on *Parcubacteria* [45], a phylum known to lack several genes responsible for the biosynthesis of essential metabolites, making it likely to be metabolically dependent on other organisms [45]. As these organisms are still largely unknown, their potential functional role remains to be clarified in future. However, our results together suggest that the introduced *Pseudomonas* bacteria may have functioned as a keystone group favouring the rare or dormant species and their associated functions in the soil [46]. The rare rhizosphere microbiome is thus probably an important and underestimated source of beneficial bacteria [46].

In addition to resource catabolism, *Pseudomonas* phytohormone production also had a significant effect, which suggest potential complex microbiome–plant feedback mediated by

hormonal signalling [20,47]. For example, it has recently been shown that the presence of certain *Variovax* bacterial strains in synthetic rhizosphere communities can restore the root growth in *Arabidopsis thaliana* via effects on bacterially produced ethylene and auxin in the rhizosphere [47]. Moreover, the presence of certain bacterial taxa can modulate the expression of scopoletin antibacterial compounds by the plant via affecting root-specific transcription factor MYB72, further shaping the assembly of rhizosphere microbiome [21]. Finally, several bacteria can play an important indirect role for pathogen suppression by boosting the plant immunity instead of having direct antagonistic effect on the pathogen, including *R. solanacearum* [48]. We found similar indirect evidence in our structured equation models: consortia-mediated effects were channelled into plant growth via effects on the resident microbiome, instead of introduction of plant-beneficial functions. Our findings are thus in line with previous findings, and suggest that probiotic inoculants could be designed to activate the functioning of existing resident microbiomes instead of bioaugmentation of communities with additional species with desired functional traits [17]. However, as our amplicon sequencing data cannot reliably distinguish the relative abundances of different inoculated *Pseudomonas* strains, or how changes in the resident community diversity and composition were linked to expression of plant growth-promoting genes, metagenomics and transcriptomics approaches and directly tailored mechanistic experiments are needed in the future. Furthermore, metabolomic and plant transcriptomics studies would be important for inferring the chemical signalling between the plant and rhizosphere microbiome [47].

## 5. Conclusion

We conclude that plant growth can be improved using species-rich inoculants that have indirect beneficial effects for the plant through compositional changes in the resident rhizosphere microbiome. We observed a positive response by rare taxa, highlighting the importance of rare biosphere for plant–microbe interactions [46]. This calls for rethinking of the traditional microbial inoculant design [8]. Instead of attempting to introduce 'plant growth-promoting' traits into the resident communities through microbial inoculants [49], we propose an alternative strategy where inoculants are designed to steer, boost and activate the resident plant growth-promoting microbes already in the rhizosphere. While more work is required to identify key bacterial taxa, their functional roles, and their chemical interplay with plants and other microbes [47], it is important to move beyond single-strain inoculants as microbe-mediated plant-beneficial effects in agricultural environments are determined by complex community-level interactions. More work is also needed to understand the extent to which the inoculant effects are determined by the composition of the resident community, and if the inoculant consortia need to be tailored specifically according to each environment. Furthermore, it is currently unclear if the inoculant effects will be transient or if they will last across plant generations. These questions could be answered by bringing together community ecology, environmental microbiology and -omics techniques to harness microbe–plant interactions for sustainable agriculture.

Data accessibility. The Miseq 16S rRNA sequencing data were deposited into the NCBI Sequence Read Archive (SRA) database under the

accession number SRP132352. Relevant data to this manuscript and script employed in the computational analyses and plotting figures are available at https://github.com/HuJamie/JieHu2021_Microbiome.

Authors' contributions. J.H.: Data curation, formal analysis, funding acquisition, investigation, methodology, validation, visualization, writing—original draft, writing—review and editing; T.Y.: funding acquisition, investigation, methodology, writing—review and editing; V.-P.F.: conceptualization, writing—review and editing; G.A.K.: supervision, writing—review and editing; Y.H.: formal analysis, writing—review and editing; M.L.: methodology; Z.W.: project administration, visualization, writing—original draft, writing—review and editing; Y.X.: funding acquisition, resources, supervision; Q.S.: resources, supervision; A.J.: funding acquisition, project administration, writing—original draft, writing—review and editing.

All authors gave final approval for publication and agreed to be held accountable for the work performed therein.

Competing interests. We declare we have no competing interests.

Funding. This research was financially supported by the National Key Research and Development Program of China (grant nos 2018YFD1000800 and SQ2021YFD1900024), National Natural Science Foundation of China (grant nos 41807045, 31972504, 41922053 and 42090060), Natural Science Foundation of Jiangsu Province (grant no. BK20180527) and the Fundamental Research Funds for the Central Universities (grant nos KY2201719; KJYQ202002; KJQN201922). A.J. and J.H. are supported by the NWO grant no. ALW.870.15.050 and the TKI top-sector grant no. KV1605 082. J.H. is supported by Chinese Scholarship Council (CSC) joint PhD scholarship (grant no. 201506850027). V.-P.F. is supported by the Royal Society (grant nos RSG\R1\180213 and CHL\R1\180031) and jointly by a grant from UKRI, Defra, and the Scottish Government, under the Strategic Priorities Fund Plant Bacterial Diseases programme (BB/T010606/1) at the University of York.

Acknowledgements. We thank Shaohua Gu and Xiaofang Wang for assisting part of the greenhouse experiment, Shanghai Biozeron Biotechnology for technical support in Miseq sequencing and Peter Veenhuizen from Utrecht University for technical support of analysing Miseq sequencing data. We also thank all the anonymous reviewers for constructive comments to improve this manuscript.

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
