## [Peer Review File · Proceedings of the Royal Society B: Biological Sciences]

Review History

RSPB-2021-1396.R0 (Original submission)

Review form: Reviewer 1

Recommendation

Accept with minor revision (please list in comments)

Scientific importance: Is the manuscript an original and important contribution to its field?

Excellent

General interest: Is the paper of sufficient general interest?

Excellent

Quality of the paper: Is the overall quality of the paper suitable?

Excellent

Is the length of the paper justified?

Yes

Should the paper be seen by a specialist statistical reviewer?

No

Do you have any concerns about statistical analyses in this paper? If so, please specify them explicitly in your report.

No

It is a condition of publication that authors make their supporting data, code and materials available - either as supplementary material or hosted in an external repository. Please rate, if applicable, the supporting data on the following criteria.

Is it accessible?

Yes

Is it clear?

Yes

Is it adequate?

Yes

Do you have any ethical concerns with this paper?

No

Comments to the Author

The authors have submitted a manuscript that presents data on how inoculating plants with a known consortia of pro-biotic bacteria can enhance the growth of the host plant. The authors claim this is through diversity and compositional changes in the rhizosphere community caused by the inoculum.

This work represents a fantastic experiments, performed thoroughly and with attention to the pitfalls of analysing complex experiments. This works gives evidence of an interesting new way of looking at the field of designed inocula, and is of great interest to a range of disciplines.

I have two major queries about the study which I did not feel were full addressed in the text, that need a bit more attention:

This study sets up a wonderful experimental design that the authors have built from some of their previous work, however, the choice of consortia consists of entirely one genera and largely of one species. Whilst I acknowledge that phenotypically they have some differences in terms of plant-promotion, would this piece not have been strengthened by the addition of other genera that were not potentially going to exclude each other? I think this is particularly important given that only two of the strain were detected in the downstream analysis (L312). I would have also thought (see minor comments) that there would be some resident Pseudomonads present therefore there is the concern of competition with the resident community that I think could be explored further. If there were more species then this may mitigate the competitive effect and therefore have a different microbiome impact. Is this possible whilst maintaining the additional functioning stated in the text? A clearer rationale for the use of these strains (beyond they are model organisms L109).

My second query is the use of a propagule of extremely high numbers (10^8 cells). The effect of this is mentioned in L370, but not expanded on. By adding that larger number of cells to the soil this will have contributed to the competitive effects of the consortia on the microbiome. In itself this is not a major problem, but I think this needs to be rationalised and accounted for.

Minor comments:

L272: I find the result in L272 at odds with my understanding of L261, and the statements at L352, L368. What do the authors mean by the term “identify effects”? If each of the strains are producing different functions (L261) then surely the presence of each strain will result in those particular functions occurring? Is this that the variance of the model is taken up entirely by the linear richness component resulting in non-significance?

L278: What were the proportions of Pseudomonas identified in the 16S sequencing of the control (no-inoculum) samples? I find it hard to believe that there were virtually no Pseudomonads in the soil at all...

L294: Could this be due to the addition of the greater number of strains and the diversity metric

used? i.e. if the authors used Bray-Curtis, then this is heavily skewed towards accounting for the most abundant organisms, which in this case would include the massive dose of 8 *Pseudomonas* strains. This is assuming they are included in this analysis...

L298-303: I think more likely the reason is an out-competing the abundant species resulting in a reduction in competitive exclusion, rather than a direct "revival" of rare species.

L377: I don't think the consortia "promoted competition with the resident community", rather I think the reduction in abundance was because of the consortia out-competed the abundant species, thereby reducing overall load. This is stated in L388.

L399: interesting hypothesis for future work and well linked with the conclusion.

Review form: Reviewer 2

Recommendation

Accept with minor revision (please list in comments)

Scientific importance: Is the manuscript an original and important contribution to its field?

Excellent

General interest: Is the paper of sufficient general interest?

Good

Quality of the paper: Is the overall quality of the paper suitable?

Excellent

Is the length of the paper justified?

Yes

Should the paper be seen by a specialist statistical reviewer?

No

Do you have any concerns about statistical analyses in this paper? If so, please specify them explicitly in your report.

No

It is a condition of publication that authors make their supporting data, code and materials available - either as supplementary material or hosted in an external repository. Please rate, if applicable, the supporting data on the following criteria.

Is it accessible?

Yes

Is it clear?

Yes

Is it adequate?

Yes

Do you have any ethical concerns with this paper?

No

Comments to the Author

This article relates to the microbial ecology of the plant rhizosphere and how to optimize plant-bacteria functions through probiotic additions. The authors' claim is that the beneficial effects of

probiotic addition are positively correlated with the species richness of the probiotic inoculum. The effects of the probiotic cocktail on plant growth were due to subsequent changes in the resident bacterial community diversity and composition, including an increase in the relative abundance of initially rare taxa. Thus the conclusion is that the probiotic additions primarily were of value to the extent that they enhanced the beneficial functions of the existing microbiota.

This is an important and interesting results even if the mechanism of plant growth enhancement is not completely understood.

However, the basic and practical implications are not entirely clear. To determine which probiotics are best in a given system, we would need to know which of the OTUs in the existing microbiota need to be facilitated and which probiotics would accomplish that purpose. This is somewhat analogous to activation a quiescent immune cell, although in this case the specific OTUs and causal links are not known. Perhaps these links could be determined in “natural” conditions, and then the insights gained could be applied to the soils degraded by agriculture. Alternatively, it might be that this type of experiment is needed for each system to tease out optimal probiotic mixtures.

It appears that the probiotics enhanced the relative abundance of rare OTUs in the existing microbiota, and these OTUs improved plant growth. From an evolutionary perspective, why wouldn't the plant-microbiota system have been selected to enhance these beneficial OTUs' abundance? What is limiting their abundance? Is their rareness due to the degraded soil conditions?

59: even if microbial species show antagonism to each other, they may function well as a probiotic consortium. If they are competing, this will induce production and secretion of antimicrobial metabolites, with the plant getting a defensive function from the microbes as a by-product of their competition. This effects would be more beneficial in environments where plant pathogens were common.

123: please expand: did consortia with more species also contain a higher total abundance of bacterial cells? i.e., was richness positively correlated with total abundance and therefore confounded?

Minor points:

90: define “all probiotic consortia”. Does this mean all combinations of consortia?

160: change to “with more details in supplementary methods”.

186: change to “More details are in supplementary methods.”

239: change to “strains”

290: how long would the change in resident microbiome last?

338: change to “these data suggest”

370: can the effects of density be separated from the effects of diversity?

414: change to “plant”

Decision letter (RSPB-2021-1396.R0)

06-Sep-2021

Dear Dr Wei:

Your manuscript has now been peer reviewed and the reviews have been assessed by an Associate Editor. The reviewers' comments (not including confidential comments to the Editor) and the comments from the Associate Editor are included at the end of this email for your reference. As you will see, the reviewers and the Editors have raised some concerns with your manuscript and we would like to invite you to revise your manuscript to address them.

Research ethics:

Use of animals and field studies:

It is a condition of publication that you make available the data and research materials supporting the results in the article. Please see our Data Sharing Policies (<https://royalsociety.org/journals/authors/author-guidelines/#data>). Datasets should be deposited in an appropriate publicly available repository and details of the associated accession number, link or DOI to the datasets must be included in the Data Accessibility section of the

article (<https://royalsociety.org/journals/ethics-policies/data-sharing-mining/>). Reference(s) to datasets should also be included in the reference list of the article with DOIs (where available).

If you wish to submit your data to Dryad (<http://datadryad.org/>) and have not already done so you can submit your data via this link [http://datadryad.org/submit?journalID=RSPB&manu=\(Document not available\)](http://datadryad.org/submit?journalID=RSPB&manu=(Document%20not%20available)), which will take you to your unique entry in the Dryad repository.

Please submit a copy of your revised paper within three weeks. If we do not hear from you within this time your manuscript will be rejected. If you are unable to meet this deadline please let us know as soon as possible, as we may be able to grant a short extension.

Best wishes,
Professor Gary Carvalho
mailto: proceedingsb@royalsociety.org

Associate Editor
Board Member: 1
Comments to Author:

Dear authors, thank you for submitting your manuscript for consideration in proceedings B. Myself and two expert reviewers have now assessed your manuscript and I'm pleased to say that we have received a very positive assessment of the work. Both reviewers have highlighted some revisions that must be addressed in a revised manuscript and I would be grateful if you could address these requested revisions in the subsequent submission.

Reviewer(s)' Comments to Author:
Referee: 1

Comments to the Author(s)

The authors have submitted a manuscript that presents data on how inoculating plants with a known consortia of pro-biotic bacteria can enhance the growth of the host plant. The authors

claim this is through diversity and compositional changes in the rhizosphere community caused by the inoculum.

This work represents a fantastic experiments, performed thoroughly and with attention to the pitfalls of analysing complex experiments. This works gives evidence of an interesting new way of looking at the field of designed inocula, and is of great interest to a range of disciplines.

I have two major queries about the study which I did not feel were full addressed in the text, that need a bit more attention:

This study sets up a wonderful experimental design that the authors have built from some of their previous work, however, the choice of consortia consists of entirely one genera and largely of one species. Whilst I acknowledge that phenotypically they have some differences in terms of plant-promotion, would this piece not have been strengthened by the addition of other genera that were not potentially going to exclude each other? I think this is particularly important given that only two of the strain were detected in the downstream analysis (L312). I would have also thought (see minor comments) that there would be some resident Pseudomonads present therefore there is the concern of competition with the resident community that I think could be explored further. If there were more species then this may mitigate the competitive effect and therefore have a different microbiome impact. Is this possible whilst maintaining the additional functioning stated in the text? A clearer rationale for the use of these strains (beyond they are model organisms L109).

My second query is the use of a propagule of extremely high numbers (10^8 cells). The effect of this is mentioned in L370, but not expanded on. By adding that larger number of cells to the soil this will have contributed to the competitive effects of the consortia on the microbiome. In itself this is not a major problem, but I think this needs to be rationalised and accounted for.

Minor comments:

L272: I find the result in L272 at odds with my understanding of L261, and the statements at L352, L368. What do the authors mean by the term “identify effects”? If each of the strains are producing different functions (L261) then surely the presence of each strain will result in those particular functions occurring? Is this that the variance of the model is taken up entirely by the linear richness component resulting in non-significance?

L278: What were the proportions of Pseudomonas identified in the 16S sequencing of the control (no-inoculum) samples? I find it hard to believe that there were virtually no Pseudomonads in the soil at all...

L294: Could this be due to the addition of the greater number of strains and the diversity metric used? i.e. if the authors used Bray-Curtis, then this is heavily skewed towards accounting for the most abundant organisms, which in this case would include the massive dose of 8 Pseudomonas strains. This is assuming they are included in this analysis...

L298-303: I think more likely the reason is an out-competing the abundant species resulting in a reduction in competitive exclusion, rather than a direct “revival” of rare species.

L377: I don't think the consortia “promoted competition with the resident community”, rather I think the reduction in abundance was because of the consortia out-competed the abundant species, thereby reducing overall load. This is stated in L388.

L399: interesting hypothesis for future work and well linked with the conclusion.

Referee: 2

Comments to the Author(s)

This article relates to the microbial ecology of the plant rhizosphere and how to optimize plant-bacteria functions through probiotic additions. The authors' claim is that the beneficial effects of probiotic addition are positively correlated with the species richness of the probiotic inoculum.

The effects of the probiotic cocktail on plant growth were due to subsequent changes in the resident bacterial community diversity and composition, including an increase in the relative abundance of initially rare taxa. Thus the conclusion is that the probiotic additions primarily were of value to the extent that they enhanced the beneficial functions of the existing microbiota.

This is an important and interesting results even if the mechanism of plant growth enhancement is not completely understood.

However, the basic and practical implications are not entirely clear. To determine which probiotics are best in a given system, we would need to know which of the OTUs in the existing microbiota need to be facilitated and which probiotics would accomplish that purpose. This is somewhat analogous to activation a quiescent immune cell, although in this case the specific OTUs and causal links are not known. Perhaps these links could be determined in “natural” conditions, and then the insights gained could be applied to the soils degraded by agriculture. Alternatively, it might be that this type of experiment is needed for each system to tease out optimal probiotic mixtures.

It appears that the probiotics enhanced the relative abundance of rare OTUs in the existing microbiota, and these OTUs improved plant growth. From an evolutionary perspective, why wouldn't the plant-microbiota system have been selected to enhance these beneficial OTUs' abundance? What is limiting their abundance? Is their rareness due to the degraded soil conditions?

59: even if microbial species show antagonism to each other, they may function well as a probiotic consortium. If they are competing, this will induce production and secretion of antimicrobial metabolites, with the plant getting a defensive function from the microbes as a by-product of their competition. This effects would be more beneficial in environments where plant pathogens were common.

123: please expand: did consortia with more species also contain a higher total abundance of bacterial cells? i.e., was richness positively correlated with total abundance and therefore confounded?

Minor points:

90: define “all probiotic consortia”. Does this mean all combinations of consortia?

160: change to “with more details in supplementary methods”.

186: change to “More details are in supplementary methods.”

239: change to “strains”

290: how long would the change in resident microbiome last?

338: change to “these data suggest”

370: can the effects of density be separated from the effects of diversity?

414: change to “plant”

Author's Response to Decision Letter for (RSPB-2021-1396.R0)

See Appendix A.

Decision letter (RSPB-2021-1396.R1)

21-Sep-2021

Dear Dr Wei

I am pleased to inform you that your manuscript entitled "Introduction of probiotic bacterial consortia promotes plant growth via impacts on the resident rhizosphere microbiome" has been accepted for publication in Proceedings B.

Data Accessibility section

Open Access

You are invited to opt for Open Access, making your freely available to all as soon as it is ready for publication under a CCBY licence. Our article processing charge for Open Access is £1700. Corresponding authors from member institutions (<http://royalsocietypublishing.org/site/librarians/allmembers.xhtml>) receive a 25% discount to these charges. For more information please visit <http://royalsocietypublishing.org/open-access>.

Paper charges

Sincerely,

Professor Gary Carvalho

Associate Editor:

Board Member

Comments to Author:

Thank you for submitting your revised manuscript. I am happy that the revised manuscript has addressed the reviewers comments.

Appendix A

Associate Editor

Board Member: 1

Comments to Author:

Dear authors, thank you for submitting your manuscript for consideration in proceedings B. Myself and two expert reviewers have now assessed your manuscript and I'm pleased to say that we have received a very positive assessment of the work. Both reviewers have highlighted some revisions that must be addressed in a revised manuscript and I would be grateful if you could address these requested revisions in the subsequent submission.

R: We thank the Editor and two anonymous reviewers for all the helpful comments and positive feedback. We have now carefully revised our manuscript accordingly and hope that our manuscript now meets the high standards of proceedings B. Please find below our detailed responses to reviewers' comments.

Reviewer(s)' Comments to Author:

Referee: 1

Comments to the Author(s)

The authors have submitted a manuscript that presents data on how inoculating plants with a known consortia of pro-biotic bacteria can enhance the growth of the host plant. The authors claim this is through diversity and compositional changes in the rhizosphere community caused by the inoculum.

This work represents a fantastic experiments, performed thoroughly and with attention to the pitfalls of analysing complex experiments. This works gives evidence of an interesting new way of looking at the field of designed inocula, and is of great interest to a range of disciplines.

R: We thank reviewer #1 for the positive feedback.

I have two major queries about the study which I did not feel were full addressed in the text, that need a bit more attention:

This study sets up a wonderful experimental design that the authors have built from some of their previous work, however, the choice of consortia consists of entirely one genera and largely of one species. Whilst I acknowledge that phenotypically they have some differences in terms of plant-promotion, would this piece not have been strengthened by the addition of other genera that were not potentially going to exclude each other? I think this is particularly important given that only two of the strain were detected in the downstream analysis (L312). I would have also thought (see minor comments) that there would be some resident Pseudomonads present therefore there is the concern of competition with the resident community that I think could be explored further. If there were more species then this may mitigate the competitive effect and therefore have a different microbiome impact. Is this possible whilst maintaining the additional functioning stated in the text? A clearer rationale for the use of these strains (beyond they are model organisms L109).

R: This is a very good question and we have now clarified the use of strains from this single genus below and in the text (on lines 108-110 of the main text). Indeed, we only selected the strains from *Pseudomonas* genus since they are model strains and already well-studied in terms of their plant growth-promotion. We fully agree that using a more diverse consortium is likely to change the results. However, we feel that this is out of scope of this study and suggest that this should be followed in the future studies.

Only two of the inoculated strains were detected in the 16S rRNA amplicon sequences and this could be partly due to used technology. The sequences we used to identify inoculated strains are based on whole 16S rRNA length, which is around 1500 bp. In contrast, 16S rRNA amplicon sequences derived from the soil total DNA contained a fragment of V4 region which was only around 500 bp in length. As a result, matching these two types of sequence data was challenging and we choose to use conservative approach and only accepted matches with 99.5% similarity as true sequence of the inoculated strains. We have now clarified this in the text on lines 154-156 of the supplementary materials and methods. We also suggest that future studies would employ metagenomics to track the inoculated strains' abundances in the rhizosphere.

We also agree that as *Pseudomonas* is one of the most abundant bacteria in soil, the competition between inoculated and resident strains of this genus is inevitable. It is possible that such induced competition could also be one of the proximate mechanisms for our results, if inoculation increases competition between the

resident *Pseudomonas* taxa which has also negative effects on the pathogen. This potential outcome is now mentioned on line 73 of main text and should be studied in more detail in the future.

My second query is the use of a propagule of extremely high numbers (10^8 cells). The effect of this is mentioned in L370, but not expanded on. By adding that larger number of cells to the soil this will have contributed to the competitive effects of the consortia on the microbiome. In itself this is not a major problem, but I think this needs to be rationalised and accounted for.

R: We have now justified this on lines 55-57 of the supplementary materials and methods. Briefly, high inoculation density was used to ensure that *Pseudomonas* consortia would have an effect when inoculated to natural soil and not go into extinction soon after inoculation.

Minor comments:

L272: I find the result in L272 at odds with my understanding of L261, and the statements at L352, L368. What do the authors mean by the term “identify effects”? If each of the strains are producing different functions (L261) then surely the presence of each strain will result in those particular functions occurring? Is this that the variance of the model is taken up entirely by the linear richness component resulting in non-significance?

R: With “Identity effect” we mean whether a presence of certain strain in consortia was significantly associated with analyzed outcomes. While it has been established that chosen *Pseudomonas* species can have various plant growth-promoting functions in varying degrees *in vitro*, they might not be expressing these functions in the soil when used as inoculant. Hence, testing the significance of species presence (*i.e.*, ‘identity effects’) both *in vitro* and *in vivo* is required for confirming this. We have now clarified this and the model we used on lines 92-93 of the main text and lines 156-163 of the supplementary methods.

L278: What were the proportions of *Pseudomonas* identified in the 16S sequencing of the control (no-inoculum) samples? I find it hard to believe that there were virtually no *Pseudomonas* in the soil at all...

R: The background *phlD* gene abundances targeting *Pseudomonas* species was below 0.001% (relative to 16S rRNA gene abundances) in the non-inoculated control treatment (Fig. S4A), confirming that *Pseudomonas* species which containing *phlD* gene were at very low abundance in the non-sterile agricultural soil used in the pot experiments. We have also noted that qPCR based on *phlD* gene only accounts for *Pseudomonas* strains that can produce 2,4-DAPG and might not capture the whole *Pseudomonas* diversity. We have now clarified this on lines 114-118 of the main text.

L294: Could this be due to the addition of the greater number of strains and the diversity metric used? *i.e.* if the authors used Bray-Curtis, then this is heavily skewed towards accounting for the most abundant organisms, which in this case would include the massive dose of 8 *Pseudomonas* strains. This is assuming they are included in this analysis...

R: As the inoculated *Pseudomonas* strains were found in relatively low abundance at the end of the experiment, they did not have disproportionately large effect on the reliability of the diversity metrics used. The metric we used for calculating alpha-diversity is based on normalised OTU abundance table, which was used to calculate phylogenetic abundance evenness (PAE), which is described on lines 129-140 of the supplementary methods as the diversity index of rhizosphere microbiome.

In Figure 3A of the main text, we used presence-absence metric to calculate beta-diversity of the resident rhizosphere microbiome to highlight the importance of rare species in the community. Qualitatively similar results ($P = 0.044$) were also obtained using other metrics (*e.g.*, Bray-Curtis distance metric suggested by reviewer #1), which we have now reported in Fig. S4C and on lines 229-230 and 300-301 of the main text.

L298-303: I think more likely the reason is an out-competing the abundant species resulting in a reduction in competitive exclusion, rather than a direct “revival” of rare species.

R: This is a good point. We have now included this as a potential explanation for the increase in the relative abundance of rare taxa (on lines 305-307 of the main text).

L377: I don’t think the consortia “promoted competition with the resident community”, rather I think the reduction in abundance was because of the consortia out-competed the abundant species, thereby reducing overall load. This is stated in L388.

R: We agree and have clarified this on lines 400-402 of the main text.

L399: interesting hypothesis for future work and well linked with the conclusion.

R: We thank reviewer#1 for the positive comment.

Referee: 2

Comments to the Author(s)

This article relates to the microbial ecology of the plant rhizosphere and how to optimize plant-bacteria functions through probiotic additions. The authors' claim is that the beneficial effects of probiotic addition are positively correlated with the species richness of the probiotic inoculum. The effects of the probiotic cocktail on plant growth were due to subsequent changes in the resident bacterial community diversity and composition, including an increase in the relative abundance of initially rare taxa. Thus the conclusion is that the probiotic additions primarily were of value to the extent that they enhanced the beneficial functions of the existing microbiota.

This is an important and interesting results even if the mechanism of plant growth enhancement is not completely understood.

R: We thank reviewer#2 for the positive feedback.

However, the basic and practical implications are not entirely clear. To determine which probiotics are best in a given system, we would need to know which of the OTUs in the existing microbiota need to be facilitated and which probiotics would accomplish that purpose. This is somewhat analogous to activation a quiescent immune cell, although in this case the specific OTUs and causal links are not known. Perhaps these links could be determined in "natural" conditions, and then the insights gained could be applied to the soils degraded by agriculture. Alternatively, it might be that this type of experiment is needed for each system to tease out optimal probiotic mixtures.

R: We fully agree that some knowledge on the existing microbial community would be required to predictably steer the functioning of existing community. This caveat is now pointed out in the discussion on lines 449-451 of the main text.

It appears that the probiotics enhanced the relative abundance of rare OTUs in the existing microbiota, and these OTUs improved plant growth. From an evolutionary perspective, why wouldn't the plant-microbiota system have been selected to enhance these beneficial OTUs' abundance? What is limiting their abundance? Is their rareness due to the degraded soil conditions?

R: This is a great question. One likely possibility is that plants and their microbes might have conflicting interests and competition between different bacteria can constrain the plant recruitment of certain (e.g., rare) taxa.

59: even if microbial species show antagonism to each other, they may function well as a probiotic consortium. If they are competing, this will induce production and secretion of antimicrobial metabolites, with the plant getting a defensive function from the microbes as a by-product of their competition. This effects would be more beneficial in environments where plant pathogens were common.

R: Indeed, we completely agree with this point that pathogen suppression could be a side effect arising from microbial competition (see lines 71-73 of the main text).

123: please expand: did consortia with more species also contain a higher total abundance of bacterial cells? i.e., was richness positively correlated with total abundance and therefore confounded?

R: No, all designed consortia contained the same total abundance of *Pseudomonas* bacteria at each richness level (defined on lines 127-129 of the main text).

Minor points:

90: define "all probiotic consortia". Does this mean all combinations of consortia?

R: Yes, this means all combinations of consortia. To make this clear, we have clarified the sentence to: "We then inoculated each combination of probiotic consortia in the tomato rhizosphere".

160: change to "with more details in supplementary methods".

R: Revised as suggested.

186: change to “More details are in supplementary methods.”

R: Revised as suggested.

239: change to “strains”

R: Revised as suggested.

290: how long would the change in resident microbiome last?

R: This is a very good question for which more experiments are required. We note this now in the discussion on lines 451-455 of the main text.

338: change to “these data suggest”

R: Revised as suggested.

370: can the effects of density be separated from the effects of diversity?

R: As the same inoculant densities were used at all consortium richness levels, this result reveals a causal link between the used inoculant richness and increase in total bacterial abundances after inoculation. However, we agree that disentangling these effects is difficult (lines 381-382 of the main text).

414: change to “plant”

R: Revised as suggested.